# Maximize to Explore: One Objective Function Fusing Estimation, Planning, and Exploration

**Zhihan Liu**[1*]    **Miao Lu**[2*]    **Wei Xiong**[3*]    **Han Zhong**[4]    **Hao Hu**[5]
**Shenao Zhang**[1]    **Sirui Zheng**[1]    **Zhuoran Yang**[6]    **Zhaoran Wang**[1]
[1]Northwestern University   [2]Stanford University   [3]University of Illinois Urbana-Champaign
[4]Peking University   [5]Tsinghua University   [6]Yale University
{zhihanliu2027,shenaozhang2028,siruizheng2025}@u.northwestern.edu
miaolu@stanford.edu, wx13@illinois.edu, hanzhong@stu.pku.edu.cn
huh22@mails.tsinghua.edu.cn, zhuoran.yang@yale.edu, zhaoranwang@gmail.com

## Abstract

In reinforcement learning (RL), balancing exploration and exploitation is crucial for achieving an optimal policy in a sample-efficient way. To this end, existing sample-efficient algorithms typically consist of three components: estimation, planning, and exploration. However, to cope with general function approximators, most of them involve impractical algorithmic components to incentivize exploration, such as data-dependent level-set constraints or complicated sampling procedures. To address this challenge, we propose an easy-to-implement RL framework called *Maximize to Explore* (MEX), which only needs to optimize *unconstrainedly* a single objective that integrates the estimation and planning components while balancing exploration and exploitation automatically. Theoretically, we prove that the MEX achieves a sublinear regret with general function approximators and is extendable to the zero-sum Markov game setting. Meanwhile, we adapt deep RL baselines to design practical versions of MEX in both the model-based and model-free settings, which outperform baselines in various MuJoCo environments with sparse reward by a stable margin. Compared with existing sample-efficient algorithms with general function approximators, MEX achieves similar sample efficiency while also enjoying a lower computational cost and is more compatible with modern deep RL methods. Our codes are available at https://github.com/agentification/MEX.

## 1  Introduction

The crux of online reinforcement learning (online RL) lies in maintaining a balance between exploiting the current knowledge of the agent about the environment and exploring unfamiliar areas [69]. To fulfill this, agents in existing sample-efficient RL algorithms predominantly undertake three tasks: i) *estimate* a hypothesis using historical data to encapsulate their understanding of the environment; ii) perform *planning* based on the estimated hypothesis to exploit their current knowledge; iii) further *explore* the unknown environment via carefully designed exploration strategies.

There exists a long line of research on integrating the aforementioned three components harmoniously, to find optimal policies in a sample-efficient manner. From theoretical perspectives, existing theories aim to minimize the notion of *online external regret* which measures the cumulative suboptimality gap of the policies learned during online learning. It is well studied that one can design both statistically and computationally efficient algorithms (e.g., upper confidence bound (UCB), [6, 39, 12, 91]) with sublinear online regret for tabular and linear Markov decision processes (MDPs). But when it

---

[*]Equal contribution.

37th Conference on Neural Information Processing Systems (NeurIPS 2023).

comes to MDPs with general function approximations, most of them involve impractical algorithmic components to incentivize exploration. Usually, to cope with general function approximations, agents need to solve constrained optimization problems within data-dependent level-sets [37, 20], or sample from complicated posterior distributions over the space of hypotheses [19, 2, 89], both of which pose considerable challenges for implementation. From a practical perspective, a prevalent approach in deep RL for balancing exploration and exploitation is to use an ensemble of neural networks [75, 59, 14, 53, 44, 18, 45], which serves as an empirical approximation of the UCB method. However, such an ensemble method suffers from high computational cost and lacks theoretical guarantee when the underly MDP is neither linear nor tabular. As for other deep RL algorithms for exploration [29, 3, 11, 9, 17, 65], such as curiosity-driven method [60], it also remains unknown in theory whether they are provably sample-efficient in the context of general function approximations.

Hence, in this paper, we are aimed at tackling these issues and answering the following question:

*Under general function approximation, can we design a sample-efficient and easy-to-implement RL framework to trade off between exploration and exploitation?*

To this end, we propose an easy-to-implement RL framework, *Maximize to Explore* (MEX), as an affirmative answer to the question. To strike a balance between exploration and exploitation, MEX propose to maximize a weighted sum of two objectives: (i) the optimal expected total return associated with a given hypothesis and (ii) the negative estimation error of that hypothesis. Consequently, MEX naturally combines planning and estimation components in just a single objective. By choosing the hypothesis that maximizes the weighted sum and executing the optimal policy with respect to the chosen hypothesis, MEX automatically balances between exploration and exploitation.

We highlight that the objective of MEX is *not* obtained by taking the Lagrange dual of the constrained optimization objective within data-dependent level-sets [37, 20, 15].This is because the coefficient of the weighted sum, which remains fixed, is data-independent and predetermined for all episodes. Contrary to Lagrangian methods, MEX does not necessitate an inner loop of optimization for dual variables, thereby circumventing the complications associated with minimax optimization. As a maximization-only framework, MEX is friendly to implementations with neural networks and does not rely on sampling or ensemble.

In the theory part, we prove that MEX achieves a sublinear regret $\widetilde{\mathcal{O}}(\mathrm{Poly}(H) \cdot d_{\mathrm{GEC}}^{1/2}(1/\sqrt{HK}) \cdot K^{1/2})$ under mild assumptions and is thus sample-efficient, where $K$ is the number of episodes and $H$ is the horizon length. Here $d_{\mathrm{GEC}}(\cdot)$ is the Generalized Eluder Coefficient (GEC) [89] that characterizes the complexity of learning the underlying MDP under general function approximations. Because the class of low-GEC MDPs includes almost all known theoretically tractable MDP instances, our proved result can be tailored to a multitude of specific settings with either a model-free or a model-based hypothesis, such as MDPs with low Bellman eluder dimension [37], MDPs of bilinear class [20], and MDPs with low witness rank [67]. Besides, thanks to the flexibility of the MEX framework, we further extend it to online RL in two-player zero-sum Markov games (MGs), for which we further extend the definition of GEC to two-player zero-sum MGs and establish the sample efficiency with general function approximations. Moving beyond theory and into practice, we adapt famous RL baselines TD3 [27] and MBPO [34] to design practical versions of MEX in model-free and model-based fashion, respectively. On various MuJoCo environments [71] with sparse rewards, experimental results show that MEX outperforms baselines steadily and significantly. Compared with other deep RL algorithms, MEX has low computational overhead and straightforward implementation while maintaining a theoretical guarantee.

**Contributions.** We conclude our contributions from three perspectives.

1. We propose an easy-to-implement RL algorithm framework MEX that *unconstrainedly* maximizes a single objective to fuse estimation and planning, automatically trading off between exploration and exploitation. Under mild structural assumptions, we prove that MEX achieves a sublinear regret $\widetilde{\mathcal{O}}(\mathrm{Poly}(H) \cdot d_{\mathrm{GEC}}^{1/2}(1/\sqrt{HK})K^{1/2})$ with general function approximators, and thus is sample-efficient. Here $d_{\mathrm{GEC}}(\cdot)$ is the generalized Eluder Coefficient (GEC) of the underlying MDP.
2. We instantiate the generic MEX framework to several model-based and model-free examples and establish corresponding theoretical results. Further, we extend the MEX framework to two-player zero-sum MGs and also prove the sample efficiency with an extended definition of GEC.
3. We design practical implementations of MEX for MDPs in both model-based and model-free styles. Experiments on various MuJoCo environments with sparse reward demonstrate the effectiveness of our proposed MEX framework.

## 1.1 Related work

**Sample-efficient RL with function approximation.** The success of DRL methods has motivated a line of work focused on function approximation scenarios. This work originated in the linear case [74, 81, 12, 39, 84, 5, 82, 57, 91, 90] and is later extended to general function approximation. Wang et al. [73] first study the general function approximation using the notion of eluder dimension [63], which takes the linear MDP [39] as a special case but with inferior results. Zanette et al. [85] consider a different type of framework based on Bellman completeness, which assumes that the class used for approximating the optimal Q-functions is closed in terms of the Bellman operator and improves the results for linear MDP. After this, Jin et al. [37] consider the eluder dimension of the class of Bellman residual associated with the RL problems, which captures more solvable problems. Another line of works focuses on the low-rank structures of the problems, where Jiang et al. [35] propose the Bellman rank for model-free RL and Sun et al. [67] propose the witness rank for model-based RL. Following these two works, Du et al. [20] propose the bilinear class, which contains more MDP models with low-rank structures [6, 67, 39, 57, 12, 91] by allowing a flexible choice of discrepancy function class. However, it is known that neither BE nor bilinear class captures each other. Dann et al. [19] first consider eluder-coefficient-type complexity measure on the Q-type model-free RL. It was later extended by Zhong et al. [89] to cover all the above-known solvable problems in both model-free and model-based manners. Foster et al. [25, 23] study another notion of complexity measure, the decision-estimation coefficient (DEC), which also unifies the Bellman eluder dimension and bilinear class. The DEC framework is appealing due to the matching lower bound in some decision-making problems, where all other complexity measures do not have. However, due to the presence of a minimax subroutine in its definition, they require a much more complicated minimax optimization oracle and cannot apply to the classical optimism-based or sampling-based methods. Chen et al. [13], Foster et al. [24] extend the vanilla DEC (to the model-free case) by incorporating an optimistic modification, which was originally referred to as the feel-good modification in Zhang [87]. Chen et al. [15] study Admissible Bellman Characterization (ABC) class to generalize BE. They also extend the GOLF algorithm and Bellman completeness in model-free RL to the model-based case by considering more general (vector-form) discrepancy loss functions to construct sharper in-sample error estimators and obtain sharper bounds compared to Sun et al. [67]. Xie et al. [79] connect the online RL with the coverage condition in the offline RL, and also study the GOLF algorithm proposed in Jin et al. [37].

**Algorithmic design in sample-efficient RL with function approximation.** The most prominent approach in this area is based on the principle of "Optimism in the Face of Uncertainty" (OFU), which dates back to Auer et al. [4]. For instance, for linear function approximation, Jin et al. [39] propose an optimistic variant of Least-Squares Value Iteration (LSVI), which achieves optimism by adding a bonus at each step. For the general case, Jiang et al. [36] first propose an elimination-based algorithm with optimism in model-free RL and is extended to model-based RL by [67]. After these, Du et al. [20], Jin et al. [37] propose two OFU-based algorithms, which are more similar to the lin-UCB algorithm [1] studied in the linear contextual bandit literature. The model-based counterpart (Optimistic Maximum Likelihood Estimation) is studied in Liu et al. [46], Chen et al. [13]. Specifically, these algorithms explicitly maintain a confidence set that contains the ground truth with high probability and conducts a constraint optimization step at each iteration to select the most optimistic hypothesis in the confidence set. The other line of work studies another powerful algorithmic framework based on posterior sampling. For instance, Zanette et al. [84] study randomized least-squares value iteration (RLSVI), which can be interpreted as a sampling-based algorithm and achieves an order-optimal result for linear MDP. For general function approximation, the works mainly follow the idea of the "feel-good" modification proposed in Zhang [87]. These algorithms start from some prior distribution over the hypothesis space and update the posterior distribution according to the collected samples but with certain optimistic modifications in either the prior or the loglikelihood function. Then the hypothesis for each iteration is sampled from the posterior and guides data collection. In particular, Dann et al. [19] studies the model-free Q-type problem, and Agarwal and Zhang [2] studies the model-based problems, but under different notions of complexity measures. Zhong et al. [89] further utilize the idea in Zhang [87] and extend the posterior sampling algorithm in Dann et al. [19] to be a unified sampling-based framework to solve both model-free and model-based RL problems, which is also shown to apply to the more challenging partially observable setting. In addition to the OFU-based algorithm and the sampling-based framework, Foster et al. [25] propose the Estimation-to-Decisions (E2D) algorithm, which can solve problems with low DEC but requires solving a complicated minimax subroutine to fit in the framework of DEC.

Due to the limitation of the page, we defer the remaining discussions of **Relationship with reward-biased maximum likelihood estimation**, **Exploration of DRL**, and **Two-player Zero-Sum Markov Game** to Appendix B.

## 2 Preliminaries

### 2.1 Episodic Markov Decision Processes and Online Reinforcement Learning

We consider an episodic MDP defined by a tuple $(\mathcal{S}, \mathcal{A}, H, \mathbb{P}, r)$, where $\mathcal{S}$ and $\mathcal{A}$ are the state and action spaces, $H$ is a finite horizon, $\mathbb{P} = \{\mathbb{P}_h\}_{h=1}^{H}$ with $\mathbb{P}_h : \mathcal{S} \times \mathcal{A} \mapsto \Delta(\mathcal{S})$ the transition kernel at the $h$-th timestep, and $r = \{r_h\}_{h=1}^{H}$ with $r_h : \mathcal{S} \times \mathcal{A} \to [0, 1]$ the reward function at the $h$-th timestep. Without loss of generality, we assume that the reward function $r_h$ is both deterministic and known.

In the episodic MDP, the agent interacts with the environment by the following *online* protocol. At the beginning of the $k$-th episode, the agent selects a policy $\pi^k = \{\pi_h^k : \mathcal{S} \mapsto \Delta(\mathcal{A})\}_{h=1}^{H}$. It takes an action $a_h^k \sim \pi_h^k(\cdot \mid x_h^k)$ at timestep $h$ and state $x_h^k$. After receiving the reward $r_h^k = r_h(x_h^k, a_h^k)$ from the environment, it transits to the next state $x_{h+1}^k \sim \mathbb{P}_h(\cdot \mid x_h^k, a_h^k)$. When the agent reaches the state $x_{H+1}^k$, it ends the $k$-th episode. Without loss of generality, we assume that the initial state $x_1 = \underline{x}$ is fixed. Our analysis can be generalized to the setting where $x_1$ is sampled from a distribution on $\mathcal{S}$.

**Policy and value functions.** For a policy $\pi = \{\pi_h : \mathcal{S} \mapsto \Delta(\mathcal{A})\}_{h=1}^{H}$, we denote by $V_h^\pi : \mathcal{S} \mapsto \mathbb{R}$ and $Q_h^\pi : \mathcal{S} \times \mathcal{A} \mapsto \mathbb{R}$ its value function and state-action value function at the $h$-th timestep, which characterizes the expected total rewards received by the agent under policy $\pi$ afterward, starting from some $x_h = x \in \mathcal{S}$ (or $x_h = x \in \mathcal{S}, a_h = a \in \mathcal{A}$, resp.), till the end of the episode. Specifically,

$$V_h^\pi(x) := \mathbb{E}_\pi \left[ \sum_{h'=h}^{H} r_{h'}(x_{h'}, a_{h'}) \,\middle|\, x_h = x \right], \quad \forall x \in \mathcal{S}, \tag{2.1}$$

$$Q_h^\pi(x, a) := \mathbb{E}_\pi \left[ \sum_{h'=h}^{H} r_{h'}(x_{h'}, a_{h'}) \,\middle|\, x_h = x, a_h = a \right], \quad \forall (x, a) \in \mathcal{S} \times \mathcal{A}. \tag{2.2}$$

We know there exists an optimal policy $\pi^*$ which has the optimal value function for all initial states [61], that is, $V_h^{\pi^*}(s) = \sup_\pi V_h^\pi(x)$ for all $h \in [H]$ and $x \in \mathcal{S}$. For simplicity, we abbreviate $V^{\pi^*}$ as $V^*$ and the optimal state-action value function $Q^{\pi^*}$ as $Q^*$. Moreover, the optimal value functions $Q^*$ and $V^*$ satisfy the following Bellman optimality equation [61], given by

$$V_h^*(x) = \max_{a \in \mathcal{A}} Q_h^*(x, a), \qquad V_{H+1}^*(x) = 0, \tag{2.3}$$

$$Q_h^*(x, a) = \left( \mathcal{T}_h Q_{h+1}^* \right)(x, a) := r_h(x, a) + \mathbb{E}_{x' \sim \mathbb{P}_h(\cdot \mid x, a)}[V_{h+1}^*(x')], \tag{2.4}$$

for all $(x, a, h) \in \mathcal{S} \times \mathcal{A} \times [H]$. We call $\mathcal{T}_h$ the Bellman optimality operator at timestep $h$. Also, for any two functions $Q_h$ and $Q_{h+1}$ on $\mathcal{S} \times \mathcal{A}$, we define

$$\mathcal{E}_h(Q_h, Q_{h+1}; x, a) := Q_h(x, a) - \mathcal{T}_h Q_{h+1}(x, a), \quad \forall (x, a) \in \mathcal{S} \times \mathcal{A}, \tag{2.5}$$

as the Bellman residual at timestep $h$ of $(Q_h, Q_{h+1})$.

**Performance metric.** We measure the online performance of an agent after $K$ episodes by *regret*. We assume that the learner predicts the optimal policy $\pi^*$ via $\pi^k$ in the $k$-th episode for each $k \in [K]$. Then the regret of $K$ episodes is defined as the cumulative suboptimality gap of $\{\pi^k\}_{k \in [K]}$[2],

$$\text{Regret}(K) = \sum_{k=1}^{K} V_1^*(x_1) - V_1^{\pi^k}(x_1). \tag{2.6}$$

The target of sample-efficient online RL is to achieve sublinear regret (2.6) with respect to $K$.

### 2.2 Function Approximation: Model-Free and Model-Based Hypothesis

To handle MDPs with large or even infinite state space, we introduce a family of function approximators. In specific, we consider a hypothesis class $\mathcal{H} = \mathcal{H}_1 \times \cdots \times \mathcal{H}_H$, which can be specified to model-based and model-free settings respectively. Also, we denote by $\Pi = \Pi_1 \times \cdots \times \Pi_H$ as the space of Markovian policies. We now specify $\mathcal{H}$ for model-free and model-based settings.

---

[2]We allow the agent to predict the optimal policy via $\pi^k$ while executing some other exploration policy $\pi_{\exp}^k$ to interact with the environment and collect data, as is considered in the related literature [67, 20, 89]

---
**Algorithm 1** Maximize to Explore (`MEX`)
---
1: **Input**: Hypothesis class $\mathcal{H}$, parameter $\eta > 0$.
2: **for** $k = 1, \cdots, K$ **do**
3:     Solve $f^k \in \mathcal{H}$ via

$$f^k = \operatorname*{argsup}_{f \in \mathcal{H}} \left\{ V_{1,f}(x_1) - \eta \cdot \sum_{h=1}^{H} L_h^{k-1}(f) \right\}. \tag{3.1}$$

4:     Execute $\pi_{\exp}(f^k)$ to collect data $\mathcal{D}^k = \{\mathcal{D}_h^k\}_{h \in [H]}$ with $\mathcal{D}_h^k = (x_h^k, a_h^k, r_h^k, x_{h+1}^k)$.
5:     Predict the optimal policy via $\pi_{f^k}$.
6: **end for**
---

**Example 2.1** (Model-free hypothesis). *For model-free hypothesis class, $\mathcal{H}$ contains state-action value functions of the MDP, i.e., $\mathcal{H}_h \subseteq \{f_h : \mathcal{S} \times \mathcal{A} \mapsto \mathbb{R}\}$. Specifically, for any $f = (f_1, \cdots, f_H) \in \mathcal{H}$, we denote $Q_f = \{Q_{h,f}\}_{h \in [H]}$ with $Q_{h,f} = f_h$. Also, we denote the corresponding optimal state-value function $V_f = \{V_{h,f}\}_{h \in [H]}$ with $V_{h,f}(\cdot) = \max_{a \in \mathcal{A}} Q_{h,f}(\cdot, a)$ and denote the corresponding optimal policy by $\pi_f = \{\pi_{h,f}\}_{h \in [H]}$ with $\pi_{h,f}(\cdot) = \arg\max_{a \in \mathcal{A}} Q_{h,f}(\cdot, a)$. Finally, we denote the optimal state-action value function under the true model, i.e., $Q^*$, by $f^*$.*

**Example 2.2** (Model-based hypothesis). *For model-based hypothesis class, $\mathcal{H}$ contains models of the MDP, i.e., the transition kernel. Specifically, we denote $f = \mathbb{P}_f = (\mathbb{P}_{1,f}, \cdots, \mathbb{P}_{H,f}) \in \mathcal{H}$. For any $(f, \pi) \in \mathcal{H} \times \Pi$, we define $V_f^\pi = \{V_{h,f}^\pi\}_{h \in [H]}$ as the state-value function induced by model $\mathbb{P}_f$ and policy $\pi$. We use $V_f = \{V_{h,f}\}_{h \in [H]}$ to denote the corresponding optimal state-value function, i.e., $V_{h,f} = \sup_{\pi \in \Pi} V_{h,f}^\pi$. The corresponding optimal policy is denoted by $\pi_f = \{\pi_{h,f}\}_{h \in [H]}$, where $\pi_{h,f} = \arg\sup_{\pi \in \Pi} V_{h,f}^\pi$. Finally, we denote the true model $\mathbb{P}$ of the MDP as $f^*$.*

We remark that the main difference between the model-based hypothesis (Example 2.2) and the model-free hypothesis (Example 2.1) is that model-based RL directly learns the transition kernel of the underlying MDP, while model-free RL learns the optimal state-action value function. Since we do not add any specific structural form to the hypothesis class, e.g., linear function or kernel function, we are in the context of *general function approximations* [67, 37, 20, 89, 15].

## 3 Algorithm: Maximize to Explore for Online RL

In this section, we propose **Maximize to Explore** (`MEX`, Algorithm 1) for online RL in MDPs with general function approximations. With a novel single objective, `MEX` automatically balances the goal of exploration and exploitation. We first give a generic algorithm framework and then instantiate it to model-free (Example 2.1) and model-based (Example 2.2) hypotheses respectively.

**Generic algorithm design.** In each episode $k \in [K]$, the agent first estimates a hypothesis $f^k \in \mathcal{H}$ using historical data $\{\mathcal{D}^s\}_{s=1}^{k-1}$ by maximizing a composite objective (3.1). Specifically, in order to achieve exploiting history knowledge while encouraging exploration, the agent considers a single objective that sums: **(a)** the negative loss $-L_h^{k-1}(f)$ induced by the hypothesis $f$, which represents the exploitation of the agent's current knowledge; **(b)** the expected total return of the optimal policy associated with this hypothesis, i.e., $V_{1,f}$, which represents exploration for a higher return. With a tuning parameter $\eta > 0$, the agent balances the weight put on the tasks of exploitation and exploration. The agent then predicts $\pi^*$ via the optimal policy with respect to the hypothesis that maximizes the composite exploration-exploitation objective function, i.e., $\pi_{f^k}$. Also, the agent executes certain exploration policy $\pi_{\exp}(f^k)$ to collect data $\mathcal{D}^k = \{(x_h^k, a_h^k, r_h^k, x_{h+1}^k)\}_{h=1}^{H}$ and updates the loss function $L_h^k(f)$. The choice of $\pi_{\exp}(f^k)$ depends on the specific MDP structure, and we refer to examples in Section 5 for detailed discussions.

We highlight that `MEX` is **not** a Lagrangian duality of constrained optimization objectives within data-dependent level-sets [37, 20, 15]. In fact, `MEX` only needs to fix the parameter $\eta$ across each episode. Thus $\eta$ is independent of data and predetermined, which contrasts Lagrangian methods that involve an inner loop of optimization for the dual variables. We also remark that we can rewrite (3.1) as a joint optimization $(f, \pi) = \arg\sup_{f \in \mathcal{H}, \pi \in \Pi} V_{1,f}^\pi(x_1) - \eta \sum_{h=1}^{H} L_h^{k-1}(f)$. When $\eta$ tends to infinity, `MEX` can be reduced to vanilla Actor-Critic framework [41], where critic $f$ minimizes estimation error and actor $\pi$ conducts greedy policy following the critic $f$. In the following two parts,

we instantiate Algorithm 1 to model-based and mode-free hypotheses by specifying the loss function $L_h^k(f)$.

**Model-free algorithm.** For model-free hypothesis (Example 2.1), (3.1) becomes

$$f^k = \underset{f \in \mathcal{H}}{\operatorname{argsup}} \left\{ \max_{a_1 \in \mathcal{A}} Q_{1,f}(x_1, a_1) - \eta \cdot \sum_{h=1}^{H} L_h^{k-1}(f) \right\}. \tag{3.2}$$

Regarding the choice of the loss function, for seek of theoretical analysis, to deal with MDPs with low Bellman eluder dimension [37] and MDPs of bilinear class [20], we assume the existence of certain function $l$ which generalizes the notion of Bellman residual.

**Assumption 3.1.** *Suppose the function $l : \mathcal{H} \times \mathcal{H}_h \times \mathcal{H}_{h+1} \times (\mathcal{S} \times \mathcal{A} \times \mathbb{R} \times \mathcal{S}) \mapsto \mathbb{R}$ satisfies:*
*i) (**Generalized Bellman completeness**) [89, 15] there exists an operator $\mathcal{P}_h : \mathcal{H}_{h+1} \mapsto \mathcal{H}_h$ such that for any $(f', f_h, f_{h+1}) \in \mathcal{H} \times \mathcal{H}_h \times \mathcal{H}_{h+1}$ and $\mathcal{D}_h = (x_h, a_h, r_h, x_{h+1}) \in \mathcal{S} \times \mathcal{A} \times \mathbb{R} \times \mathcal{S}$, it holds*

$$l_{f'}\big((f_h, f_{h+1}), \mathcal{D}_h\big) - l_{f'}\big((\mathcal{P}_h f_{h+1}, f_{h+1}), \mathcal{D}_h\big) = \mathbb{E}_{x_{h+1} \sim \mathbb{P}_h(\cdot|x_h, a_h)}\big[l_{f'}\big((f_h, f_{h+1}), \mathcal{D}_h\big)\big],$$

*where we require that $\mathcal{P}_h f_{h+1}^* = f_h^*$ and that $\mathcal{P}_h f_{h+1} \in \mathcal{H}_h$ for any $f_{h+1} \in \mathcal{H}_{h+1}$ and step $h \in [H]$;*
*ii) (**Boundedness**) it holds that $\sup_{f' \in \mathcal{H}} \|l_{f'}((f_h, f_{h+1}), \mathcal{D}_h)\|_\infty \leq B_l$ for some constant $B_l > 0$.*

We then set the loss function $L_h^k$ as an empirical estimation of the generalized squared Bellman error $|\mathbb{E}_{x_{h+1} \sim \mathbb{P}_h(\cdot|x_h, a_h)}[l_{f^s}((f_h, f_{h+1}), \mathcal{D}_h^s)]|^2$, given by

$$L_h^k(f) = \sum_{s=1}^{k} l_{f^s}\big((f_h, f_{h+1}), \mathcal{D}_h^s\big)^2 - \inf_{f_h' \in \mathcal{H}_h} \sum_{s=1}^{k} l_{f^s}\big((f_h', f_{h+1}), \mathcal{D}_h^s\big)^2. \tag{3.3}$$

We remark that the subtracted infimum term in (3.3) is to handle the variance terms in the estimation to achieve a fast theoretical rate. Similar essential ideas are also adopted by [37, 78, 19, 38, 52, 2, 89, 15].

**Model-based algorithm.** For model-based hypothesis (Example 2.2), (3.1) becomes

$$f^k = \underset{f \in \mathcal{H}}{\operatorname{argsup}} \left\{ \sup_{\pi \in \Pi} V_{1, \mathbb{P}_f}^\pi(x_1) - \eta \sum_{h=1}^{H} L_h^{k-1}(f) \right\}, \tag{3.4}$$

which is a joint optimization over the model $\mathbb{P}_f$ and the policy $\pi$. In the model-based algorithm, we choose the loss function $L_h^k$ as the negative log-likelihood loss, defined as

$$L_h^k(f) = -\sum_{s=1}^{k} \log \mathbb{P}_{h,f}(x_{h+1}^s | x_h^s, a_h^s). \tag{3.5}$$

# 4   Regret Analysis for MEX Framework

In this section, we establish a regret analysis for the MEX framework (Algorithm 1). We give a generic theoretical guarantee which holds for both model-free and model-based settings. We first present three key assumptions needed for sample-efficient learning with MEX. In Section 5, we specify the generic theory to specific examples of MDPs and hypothesis classes satisfying these assumptions.

Firstly, we assume that the hypothesis class $\mathcal{H}$ is well-specified, containing the true hypothesis $f^*$.

**Assumption 4.1** (Realizablity). *We assume that the true hypothesis $f^* \in \mathcal{H}$.*

Then we need to make a structural assumption on the MDP to ensure sample-efficient online learning. Inspired by Zhong et al. [89], we require the MDP to have low *Generalized Eluder Coefficient* (GEC). A low GEC indicates that the agent can effectively mitigate out-of-sample prediction errors by minimizing in-sample errors derived from historical data. To introduce, we define a discrepancy function $\ell_{f'}(f; \xi_h) : \mathcal{H} \times \mathcal{H} \times (\mathcal{S} \times \mathcal{A} \times \mathbb{R} \times \mathcal{S}) \mapsto \mathbb{R}$ which characterizes the error of a hypothesis $f \in \mathcal{H}$ on data $\xi_h = (x_h, a_h, r_h, x_{h+1})$. Specific choices of $\ell$ are given in Section 5 for concrete model-free and model-based examples.

**Assumption 4.2** (Low Generalized Eluder Coefficient [89]). *We assume that given an $\epsilon > 0$, there exists $d(\epsilon) \in \mathbb{R}_+$, such that for any sequence of $\{f^k\}_{k \in [K]} \subset \mathcal{H}$, $\{\pi_{\exp}(f^k)\}_{k \in [K]} \subset \Pi$,*

$$\sum_{k=1}^{K} V_{1, f^k} - V_1^{\pi_{f^k}} \leq \inf_{\mu > 0} \left\{ \frac{\mu}{2} \sum_{h=1}^{H} \sum_{k=1}^{K} \sum_{s=1}^{k-1} \mathbb{E}_{\xi_h \sim \pi_{\exp}(f^s)}[\ell_{f^s}(f^k; \xi_h)] + \frac{d(\epsilon)}{2\mu} + \sqrt{d(\epsilon)HK} + \epsilon HK \right\}.$$

*We denote the smallest number $d(\epsilon) \in \mathbb{R}_+$ satisfying this condition as $d_{\mathrm{GEC}}(\epsilon)$.*

As is shown by Zhong et al. [89], the low-GEC MDP class covers almost all known theoretically tractable MDP instances, such as linear MDP [81, 39], linear mixture MDP [5, 57, 12], MDPs of low witness rank [67], MDPs of low Bellman eluder dimension [37], and MDPs of bilinear class [20].

Finally, we make a concentration-style assumption which characterizes how the loss function $L_h^k$ is related to the expectation of the discrepancy function $\mathbb{E}[\ell]$ appearing in the definition of GEC. For ease of presentation, we assume that $\mathcal{H}$ is finite, i.e., $|\mathcal{H}| < \infty$, but our result can be directly extended to an infinite $\mathcal{H}$ using covering number arguments [72, 37, 49, 38].

**Assumption 4.3** (Generalization). *We assume that $\mathcal{H}$ is finite, i.e., $|\mathcal{H}| < +\infty$, and that with probability at least $1 - \delta$, for any episode $k \in [K]$ and hypothesis $f \in \mathcal{H}$, it holds that*

$$\sum_{h=1}^{H} L_h^{k-1}(f^*) - L_h^{k-1}(f) \lesssim -\sum_{h=1}^{H}\sum_{s=1}^{k-1}\mathbb{E}_{\xi_h \sim \pi_{\exp}(f^s)}[\ell_{f^s}(f;\xi_h)] + B\big(H\log(HK/\delta) + \log(|\mathcal{H}|)\big),$$

*where $B = B_l^2$ for model-free hypothesis (Assumption 3.1) and $B = 1$ for model-based hypothesis.*

Such a concentration style inequality is well known in the literature of online RL with general function approximation and similar analysis is also adopted by [37, 15]. With Assumptions 4.1, 4.2, and 4.3, we can present our main result (see Appendix D.1 for a proof).

**Theorem 4.4** (Online regret of MEX (Algorithm 1)). *Under Assumptions 4.1, 4.2, and 4.3, by setting*

$$\eta = \sqrt{\frac{d_{\mathrm{GEC}}(1/\sqrt{HK})}{(H\log(HK/\delta) + \log(|\mathcal{H}|)) \cdot B \cdot K}},$$

*then the regret of Algorithm 1 after $K$ episodes is upper bounded by*

$$\mathrm{Regret}(K) \lesssim \sqrt{d_{\mathrm{GEC}}(1/\sqrt{HK}) \cdot (H\log(HK/\delta) + \log(|\mathcal{H}|)) \cdot B \cdot K},$$

*with probability at least $1 - \delta$. Here $d_{\mathrm{GEC}}(\cdot)$ is defined in Assumption 4.2.*

Theorem 4.4 shows that the regret of Algorithm 1 scales with the square root of the number of episodes $K$ and the polynomials of horizon $H$, GEC $d_{\mathrm{GEC}}(1/\sqrt{K})$, and log covering number $\log\mathcal{N}(\mathcal{H}, 1/K)$. When the number of episodes $K$ tends to infinity, the average regret vanishes, meaning that the output policy of Algorithm 1 achieves global optimality. Since the regret of Algorithm 1 is sublinear with respect to the number of episodes $K$, Algorithm 1 is proved to be sample-efficient. In Appendix C, we extend the algorithm framework and the analysis to the two-player zero-sum Markov game (MG) setting, for which we also extend the definition of GEC to two-player zero-sum MGs.

Besides, as we can see from Theorem 4.4 and its specifications in Section 5, MEX matches existing theoretical results in the literature of online RL with general function approximations [89, 67, 20, 37, 19, 2]. But in the meanwhile, MEX does not require explicitly solving a constrained optimization problem within data-dependent level-sets or performing a complex sampling procedure. This advantage makes MEX a principled approach with easier practical implementation. We conduct deep RL experiments for MEX in Section 6 to demonstrate its power in complicated online problems.

## 5 Examples of MEX Framework

In this section, we specify Algorithm 1 to model-based and model-free hypothesis classes for various examples of MDPs of low GEC (Assumption 4.2), including MDPs with low witness rank [67], MDPs with low Bellman eluder dimension [37], and MDPs of bilinear class [20]. For ease of presentation, we assume that $|\mathcal{H}| < \infty$, but our result can be directly extended to infinite $\mathcal{H}$ using covering number arguments [72, 37, 49]. All the proofs of the propositions in this section are in Appendix E.

We note that another important step in specifying Theorem 4.4 to concrete hypothesis classes is to check Assumption 4.3 (supervised learning guarantee). It is worth highlighting that, in our analysis, for both model-free and model-based hypotheses, we provide supervised learning guarantees in a neat and unified manner, independent of specific MDP structures.

### 5.1 Model-free online RL in Markov Decision Processes

In this subsection, we specify Algorithm 1 for model-free hypothesis class $\mathcal{H}$ (Example 2.1). For a model-free hypothesis class, we choose the discrepancy function $\ell$ as, given $\mathcal{D}_h = (x_h, a_h, r_h, x_{h+1})$,

$$\ell_{f'}(f; \mathcal{D}_h) = \big(\mathbb{E}_{x_{h+1} \sim \mathbb{P}_h(\cdot|x_h,a_h)}[l_{f'}((f_h, f_{h+1}), \mathcal{D}_h)]\big)^2. \tag{5.1}$$

where the function $l : \mathcal{H} \times \mathcal{H}_h \times \mathcal{H}_{h+1} \times (\mathcal{S} \times \mathcal{A} \times \mathbb{R} \times \mathcal{S}) \mapsto \mathbb{R}$ satisfies Assumption 3.1. We specify the choice of $l$ in concrete examples of MDPs later. In the following, we check and specify Assumptions 4.2 and 4.3 in Section 4 for model-free hypothesis classes.

**Proposition 5.1** (Generalization: model-free RL). *We assume that $\mathcal{H}$ is finite, i.e., $|\mathcal{H}| < +\infty$. Then under Assumption 3.1, with probability at least $1 - \delta$, for any $k \in [K]$ and $f \in \mathcal{H}$, it holds that*

$$\sum_{h=1}^{H} L_h^{k-1}(f^*) - L_h^{k-1}(f) \lesssim -\sum_{h=1}^{H}\sum_{s=1}^{k-1} \mathbb{E}_{\xi_h \sim \pi_{\exp}(f^s)}[\ell_{f^s}(f;\xi_h)] + B_l^2\big(H\log(HK/\delta) + \log(|\mathcal{H}|)\big),$$

*where $L$ and $\ell$ are defined in (3.3) and (5.1) respectively. Here $B_l$ is specified in Assumption 3.1.*

Proposition 5.1 specifies Assumption 4.3 for model-free hypothesis classes. For Assumption 4.2, we need structural assumptions on the MDP. Given an MDP with generalized eluder dimension $d_{\mathrm{GEC}}$, we have the following corollary of our main theoretical result (Theorem 4.4).

**Corollary 5.2** (Online regret of MEX: model-free hypothesis). *Given an MDP with generalized eluder coefficient $d_{\mathrm{GEC}}(\cdot)$ and a finite model-free hypothesis class $\mathcal{H}$ with $f^* \in \mathcal{H}$, under Assumption 3.1, setting*

$$\eta = \sqrt{\frac{d_{\mathrm{GEC}}(1/\sqrt{HK})}{(H\log(HK/\delta) + \log(|\mathcal{H}|)) \cdot B_l^2 \cdot K}}, \tag{5.2}$$

*then the regret of Algorithm 1 after $K$ episodes is upper bounded by*

$$\mathrm{Regret}(T) \lesssim B_l \cdot \sqrt{d_{\mathrm{GEC}}(1/\sqrt{HK}) \cdot (H\log(HK/\delta) + \log(|\mathcal{H}|)) \cdot K}, \tag{5.3}$$

*with probability at least $1 - \delta$. Here $B_l$ is specified in Assumption 3.1.*

Corollary 5.2 can be directly specified to MDPs with low GEC, including MDPs with low Bellman eluder dimension [37] and MDPs of bilinear class [20]. See Appendix E.1 for a detailed discussion.

## 5.2 Model-based online RL in Markov Decision Processes

In this subsection, we specify Algorithm 1 for model-based hypothesis class $\mathcal{H}$ (Example 2.2). For model-based hypothesis class, we choose the discrepancy function $\ell$ in Assumption 4.2 and 4.3 as the hellinger distance. Given data $\mathcal{D}_h = (x_h, a_h, r_h, x_{h+1})$, we let

$$\ell_{f'}(f;\mathcal{D}_h) = D_{\mathrm{H}}(\mathbb{P}_{h,f}(\cdot|x_h,a_h)\|\mathbb{P}_{h,f^*}(\cdot|x_h,a_h)), \tag{5.4}$$

where $D_{\mathrm{H}}(\cdot\|\cdot)$ denotes the Hellinger distance. We note that by (5.4), the discrepancy function $\ell$ does not depend on the input $f' \in \mathcal{H}$. In the following, we check and specify Assumption 4.2 and 4.3.

**Proposition 5.3** (Generalization: model-based RL). *We assume that $\mathcal{H}$ is finite, i.e., $|\mathcal{H}| < +\infty$. Then with probability at least $1 - \delta$, for any $k \in [K]$, $f \in \mathcal{H}$, it holds that*

$$\sum_{h=1}^{H} L_h^{k-1}(f^*) - L_h^{k-1}(f) \lesssim -\sum_{h=1}^{H}\sum_{s=1}^{k-1} \mathbb{E}_{\xi_h \sim \pi_{\exp}(f^s)}[\ell_{f^s}(f;\xi_h)] + H\log(H/\delta) + \log(|\mathcal{H}|),$$

*where $L$ and $\ell$ are defined in (3.5) and (5.4) respectively.*

Proposition 5.3 specifies Assumption 4.3 for model-based hypothesis classes. For Assumption 4.2, we also need structural assumptions on the MDP. Given an MDP with generalized eluder dimension $d_{\mathrm{GEC}}$, we have the following corollary of our main theoretical result (Theorem 4.4).

**Corollary 5.4** (Online regret of MEX: model-based hypothesis). *Given an MDP with generalized eluder coefficient $d_{\mathrm{GEC}}(\cdot)$ and a finite model-based hypothesis class $\mathcal{H}$ with $f^* \in \mathcal{H}$, by setting*

$$\eta = \sqrt{\frac{d_{\mathrm{GEC}}(1/\sqrt{HK})}{(H\log(H/\delta) + \log(|\mathcal{H}|)) \cdot K}}, $$

*then the regret of Algorithm 1 after $K$ episodes is upper bounded by, with probability at least $1 - \delta$,*

$$\mathrm{Regret}(K) \lesssim \sqrt{d_{\mathrm{GEC}}(1/\sqrt{HK}) \cdot (H\log(H/\delta) + \log(|\mathcal{H}|)) \cdot K}, \tag{5.5}$$

Corollary 5.4 can be directly specified to MDPs with low GEC, including MDPs with low witness rank [67]. We refer to Appendix E.2 for a detailed discussion.

# 6 Experiments

In this section, we aim to answer the following two questions: **(a)** What are the practical approaches to implementing `MEX` in both model-based (`MEX-MB`) and model-free (`MEX-MF`) settings? **(b)** Can `MEX` handle challenging exploration tasks, especially in sparse reward scenarios?

**Experimental setups.** We evaluate the effectiveness of `MEX` by assessing its performance in both standard gym locomotion tasks and sparse reward locomotion and navigation tasks within the MuJoCo [71] environment. For sparse reward tasks, we select `cheetah-vel`, `walker-vel`, `hopper-vel`, `ant-vel`, and `ant-goal` adapted from Yu et al. [83], where the agent receives a reward only when it successfully attains the desired velocity or goal. To adapt to deep RL settings, we consider infinite-horizon $\gamma$-discounted MDPs and `MEX` variants. We report the results averaged over five random seeds. The full experimental settings are in Appendix H.

**Implementation details.** For the model-based variant `MEX-MB`, we use the following objective:

$$\max_\phi \max_\pi \mathbb{E}_{(x,a,r,x')\sim\beta}\left[\log\mathbb{P}_\phi(x',r\,|\,x,a)\right] + \eta'\cdot\mathbb{E}_{x\sim\sigma}\left[V^\pi_{\mathbb{P}_\phi}(x)\right], \tag{6.1}$$

where we denote by $\sigma$ the initial state distribution, $\beta$ the replay buffer, and $\eta'$ corresponds to $1/\eta$ in the previous theory sections. We leverage the *score function* to obtain the model value gradient $\nabla_\phi V^\pi_{\mathbb{P}_\phi}$ in a similar way to likelihood ratio policy gradient [70], with the gradient of action log-likelihood replaced by the gradient of state and reward log-likelihood in the model. Specifically,

$$\nabla_\phi \mathbb{E}_{x\sim\sigma}\left[V^\pi_{\mathbb{P}_\phi}(x)\right] = \mathbb{E}_{\tau^\pi_\phi}\left[\left(r + \gamma V^\pi_{\mathbb{P}_\phi}(x') - Q^\pi_{\mathbb{P}_\phi}(x,a)\right)\cdot\nabla_\phi\log\mathbb{P}_\phi(x',r\,|\,x,a)\right], \tag{6.2}$$

where $\tau^\pi_\phi$ is the trajectory under policy $\pi$ and transition $\mathbb{P}_\phi$, starting from $\sigma$. We refer the readers to previous works [62, 76] for a derivation of (6.2). The model $\phi$ and policy $\pi$ in (6.1) are updated iteratively in a `Dyna` [68] style, where model-free policy updates are performed on model-generated data. Particularly, we adopt `SAC` [30] to update the policy $\pi$ and estimate the value $Q^\pi_{\mathbb{P}_\phi}$ by performing temporal difference on the model data generated by $\mathbb{P}_\phi$. We also follow [62] to update the model using mini-batches from $\beta$ and normalize the advantage $r_h + \gamma V^\pi_{\mathbb{P}_\phi} - Q^\pi_{\mathbb{P}_\phi}$ within each mini-batch.

For the model-free variant `MEX-MF`, we observe from (3.2) that adding a maximization bias term to the standard TD error is sufficient for exploration. However, this may lead to instabilities as the bias term only involves the state-action value function of the current policy, and thus the policy may be ever-changing. To address this issue, we adopt a similar treatment as in `CQL` [42] by subtracting a baseline state-action value from random policy $\mu = \text{Unif}(\mathcal{A})$ and obtain the following objective:

$$\max_\theta \max_\pi \mathbb{E}_\beta\left[\left(r + \gamma Q_\theta(x',a') - Q_\theta(x,a)\right)^2\right] + \eta'\cdot\mathbb{E}_\beta\left[\mathbb{E}_{a\sim\pi}Q_\theta(x,a) - \mathbb{E}_{a\sim\mu}Q_\theta(x,a)\right]. \tag{6.3}$$

We update $\theta$ and $\pi$ in (6.3) iteratively in an actor-critic fashion. Due to space limits, we refer the readers to Appendix H for more implementation details of `MEX-MF`.

**Results.** We report the performance of `MEX-MB` and `MEX-MF` in Figures 1 and 2, respectively. We compare `MEX-MB` with `MBPO` [34], where our method differs from `MBPO` *only* in the inclusion of the value gradient in (6.2) during model updates. We find that `MEX-MB` offers an easy implementation with minimal computational overhead and yet remains highly effective across sparse and standard MuJoCo tasks. Notably, in the sparse reward settings, `MEX-MB` excels at achieving the goal velocity and outperforms `MBPO` by a stable margin. In standard gym tasks, `MEX-MB` showcases greater sample efficiency in challenging high-dimensional tasks with higher asymptotic returns.

We then compare `MEX-MF` with the model-free baseline `TD3` [27]. We observe that `TD3` fails in many sparse reward tasks, while `MEX-MF` significantly boosts the performance. In standard MuJoCo gym tasks, `MEX-MF` also steadily outperforms `TD3` with faster convergence and higher final returns.

# 7 Conclusions

In this paper, we introduce a novel RL algorithm framework—*Maximize to Explore* (`MEX`)—aimed at striking a balance between exploration and exploitation in online learning scenarios. `MEX` is provably sample-efficient under general function approximations and is easy to implement. Theoretically, we prove that under mild structural assumptions (low generalized eluder coefficient (GEC)), `MEX` achieves

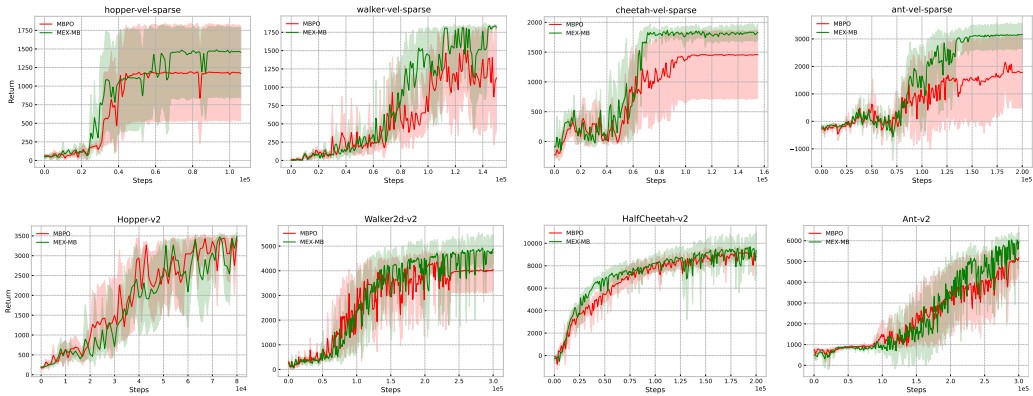

Figure 1: Model-based `MEX-MB` in sparse and standard MuJoCo locomotion tasks.

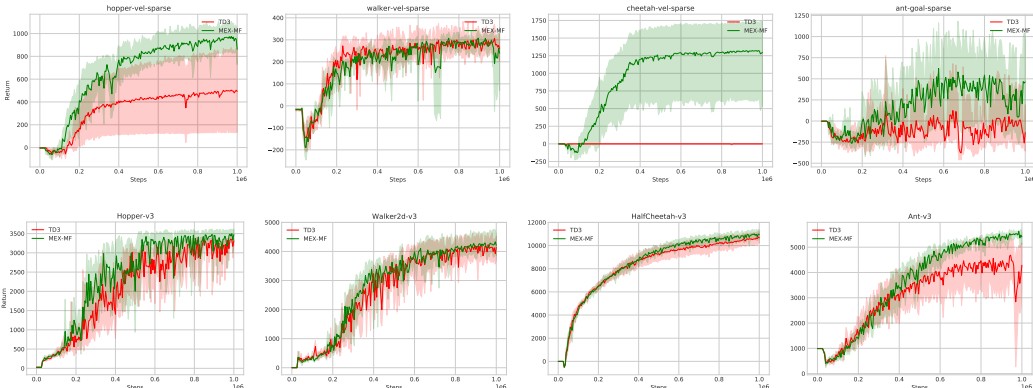

Figure 2: Model-free `MEX-MF` in sparse and standard MuJoCo locomotion tasks.

$\widetilde{\mathcal{O}}(\sqrt{K})$-online regret for MDPs. We further extend the definition of GEC and `MEX` framework to two-player zero-sum Markov games (see Appendix C) and also prove the $\widetilde{\mathcal{O}}(\sqrt{K})$-online regret. In practice, we adapt `MEX` to deep RL methods in both model-based and model-free styles and apply them to sparse-reward MuJoCo environments, outperforming baselines significantly. We hope our work can shed light on future research of designing both statistically efficient and practically effective RL algorithms with powerful function approximation.

## Acknowledgement

Zhaoran Wang acknowledges National Science Foundation (Awards 2048075, 2008827, 2015568, 1934931), Simons Institute (Theory of Reinforcement Learning), Amazon, J.P. Morgan, and Two Sigma for their supports.

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
