# OpenReview forum: "Maximize to Explore: One Objective Function Fusing Estimation, Planning, and Exploration"
_NeurIPS.cc/2023/Conference — NeurIPS 2023 spotlight_

### Official Review · Reviewer_J9HB · 2023-06-10

**Soundness:** 3 good
**Presentation:** 3 good
**Contribution:** 3 good
**Rating:** 6
**Confidence:** 3

**Summary:**

This paper studied reinforcement learning with general function approximation setting. The authors proposed a new MEX framework, which unifies the exploration and exploitation within a unconstrained optimization objective. Under the structural assumptions about low GEC, they established regret upper bound for learning with their framework. Moreover, they conducted experiments in Mujoco under both model-based and model-free setting and achieved promising results.

**Strengths:**

Computational efficiency is indeed an crucial shortage of existing provably efficient algorithm frameworks. I'm glad to see some effort to investigate how to close those gaps.

The idea is clean, the proof is sound to me. The experiments look interesting and promising.

**Weaknesses:**

(1) For the model-free setting, the objective (Eq. 3.1) may not be as easily to optimize as it looks like. By Eq. 3.3, the definition of L_h^k, solving Eq. 3.1 requires to solving a mini-max optimization problem.
On the other hand, those constrained optimization objectives in previous literatures (like [1] and [2]) can also be directly converted to a Lagrangian form, which can also be regarded as one objective (a with minimax optimization problem).
I don't think it is easy to conclude the objective in this paper is indeed easier to implement than the others.

(2) There is not much technique novelty. The proof techniques can be found in previous literatures. The maximization objective is quite straightforward, which just convert the confidence interval constrained optimization objective in previous literatures to a Lagrangian style objective, although there are indeed some difference (here the dual parameter is fixed).


[1] Jin et. al., Bellman Eluder Dimension: New Rich Classes of RL Problems, and Sample-Efficient Algorithms

[2] Du et. al., Bilinear Classes: A Structural Framework for Provable Generalization in RL

**Questions:**

(1) for the model-free setting, does there exist a general choice of $l$ function in Assumption 3.1? When Assumption 4.2 is satisfied, does $l$ function always exist?

**Limitations:**

N.A.

---

> ### Author Rebuttal · Authors · 2023-08-08
>
> **Weakness 1: For the model-free setting, the objective (3.1) may not be as easy to optimize as it looks like. By (3.3), the definition of L_h^k, solving (3.1) requires to solving a mini-max optimization problem. On the other hand, those constrained optimization objectives in previous literature (like [1, 2]) can also be directly converted to a Lagrangian form, which can also be regarded as one objective (with a minimax optimization problem). I don't think it is easy to conclude the objective in this paper is indeed easier to implement than the others.**
>
> Firstly, we highlight that MEX is a maximization objective as a whole, w.r.t. value + loss. In contrast, using the Lagrangian form to solve previous constrained optimization program gives a minimax objective w.r.t. value + dual * loss, which is indeed harder to solve than MEX. If coupled with the choice of $L_h^k$ in (3.3) [1], their target would be even harder than minimax optimization.
>
> Secondly, we need to clarify that the definition of $L_h^k$ in (3.3) is only a specific estimator construction in theory. It is indeed used to handle the double-sampling issue to achieve a sharper supervised guarantee (Proposition 5.1). Under Bellman completeness assumption (Assumption 3.1), the extra infimum term can help us to control the variance term involved in estimating things like $(E[\mathcal{E}(f)])^2$. Please refer to **General Response Q1** for more explanation for this. On the other hand, another straightforward choice of $L_h^k$ is to use the sample mean with $m$ trajectories as the estimator to achieve a low variance at the cost of a worse estimation guarantee (see [2] for a similar algorithmic design). From this perspective, the infimum term in (3.3) is not necessary, and solving (3.1) only involves maximization instead of minimax optimization.
>
> Finally, in practice, we can simply choose $L_h^k$ to be the squared TD error as an approximation, which gives a pure maximization objective. Also, for the model-based case, a simple log-likelihood estimator (3.5) is enough, which also doesn't involve minimax optimization.
>
> **Weakness 2: There is not much technique novelty. The maximization objective is quite straightforward, which just convert the confidence interval constrained optimization objective in previous literatures to a Lagrangian style objective, although there are indeed some difference (here the dual parameter is fixed).**
>
> We clarify our technical novelty in the following.
>
> As discussed in our paper, MEX is not simply a lagrangian form of the optimistic planning method proposed by existing works, e.g., [1, 2]. The theoretical gap is also non-trivial. Firstly, using Lagrangian duality to transform the constrained optimization to a minimax optimization requires both the objective function and the constraints to be convex, which in general does not hold for these optimistic planning methods. Secondly, analysis techniques in the previous papers with optimistic planning can **NOT** be generalized to our algorithm, and new proof techniques are in need.
>
> To illustrate this, we point out that in the regret decomposition (Line 871), we make a tighter analysis of the value difference $V^\star -V_{f^k}$ (Term (i)). In traditional optimistic planning papers, they need the selected hypothesis $f^k$ to maximize $V_f$ s.t. $L(f)\le \beta$ for some $\beta$. As concentration analysis tells that $L(f^\star)\le \beta$, they simply upper bound the model value difference $V^\star -V_{f^k} = V_{f^{\star}} - V_{f^k}$  by zero.
>
> While in MEX, we do not solve such a constrained optimization problem. Hence we are not proving that $V^\star -V_{f^k}\le 0$ for the selected hypothesis $f^k$. In contrast, our MEX objective (3.1) shows that its upper bound can be further refined as $-L(f^k) +L(f^\star)$. It turns out that this term can cancel some parts in the upper bound of the other value difference $V_{f^k} - V^{\pi_{f^k}}$ (Term (ii) in regret decomposition). The key here is to show the concentration type inequality (Assumption 4.3). After this cancellation, the remaining parts of both Term (i) and Term (ii) are relatively easy to handle. In contrast, traditional optimistic planning papers consider to upper bound the whole Term (ii) $V_{f^k} - V^{\pi_{f^k}}$, which relies on more complicated arguments. Hence our proof technique is novel compared with previous literatures.
>
> **Question 1: for the model-free setting, does there exist a general choice of $l$ function in Assumption 3.1? When Assumption 4.2 is satisfied, does $l$ function always exist?**
>
> Firstly, we note that the introduction of the abstract function $l$ in Assumption 3.1 is mainly due to the technical consideration of our paper, since we aim to cover all existing known theoretically tractable model-free MDP instances. E.g., MDPs of bilinear class [2], which itself is defined based upon the abstract function $l$ (Line 944). Indeed, for most of the time, the loss choice is natural in practice as the TD error $f_h(x_h,a_h) - r_h - f_{h+1}(x_{h+1})$.
>
> Secondly, the existence of such an $l$ should actually be regarded as a part of Assumption 4.2 (low GEC) for the model-free setting, since the discrepancy function $\ell$ in Assumption 4.2 is chosen as $\ell_f = (E_{x_{h+1}\sim P_h(\cdot|x_h,a_h)}[l_f(x_h,a_h,x_{h+1})])^2$ (see Eq. 5.1). In Section 5 and Appendix E, we clarify the choice of $l$ for the specific MDP instances we consider, which satisfies Assumption 3.1 and 4.2 simultaneously. This demonstrates the existance of the desired function $l$. Thanks for your question and we will make it clearer in the revision.
>
>
> [1] Jin, Chi, Qinghua Liu, and Sobhan Miryoosefi. "Bellman eluder dimension: New rich classes of rl problems, and sample-efficient algorithms." Advances in neural information processing systems 34 (2021): 13406-13418.
>
> [2] Du, Simon, et al. "Bilinear classes: A structural framework for provable generalization in rl." International Conference on Machine Learning. PMLR, 2021.

---

> > ### Comment · Reviewer_J9HB · 2023-08-10
> >
> > Thanks for the response by authors. My concerns are addressed and I would like to keep the score for acceptance.

---

> > > ### Author Response · Authors · 2023-08-10
> > > **Reply by Authors**
> > >
> > > Dear Reviewer J9HB,
> > >
> > > Thank you for your review and support. We will incorporate your valuable suggestions into our paper as we revise it based on the feedback from all reviewers. Your comments greatly assist us in strengthening the overall quality of our work.
> > >
> > > Best regards,
> > > Authors

---

### Official Review · Reviewer_wpGc · 2023-06-24

**Soundness:** 3 good
**Presentation:** 3 good
**Contribution:** 3 good
**Rating:** 7
**Confidence:** 2

**Summary:**

This theory paper proposes an algorithmic framework where the "hypothesis" (model for MB methods, Q function for MF methods) for each iteration is chosen by maximizing one objective: the hypothesis likelihood (i.e., NLL for MB methods, TD error for MF methods) plus the expected returns (value function at the initial state). The paper provides regret guarantees matching prior work. The paper proposes a practical variant of the method (building on top of MBPO and TD3) that shows some promise on sparse versions of benchmark continuous control tasks (while matching performance on the standard, non-sparse versions).

**Strengths:**

**Originality**: To the best of my knowledge the proposed frame work is original.
* One prior paper that's related to the model-based version of this paper is https://arxiv.org/abs/2110.02758. This that paper lacks the theoretical contributions of this paper, and comes at the problem from a different perspective, the resulting algorithms look surprisingly similar: both learn a model that is optimized to (1) have high likelihood and (2) produce high-return policies. I'd be curious to see if the current paper also performs well on the stochastic gridworlds used in the prior paper to showcase the benefits of the "optimistic" model.

**Clarity**: I generally found the writing clear, which is especially impressive for a large theory project packed into 9 pages. The bottom of this text block has a few suggestions for improving clarity. A few high-level things
* The related work section is generally well written, including a pretty thorough review of prior work. One suggestion is to explain the differences from prior work. What are the limitations that this paper will address?
* I didn't get much from reading Section 5; I'd prefer that the space be spent on more intuition for the previous section, or more empirical results.


**Significance**: Overall, it seems like the proposed framework is a novel and potentially useful way of designing better RL algorithms. I especially appreciate the simplicity of the proposed approach. I think that the paper may be a bit closer to prior work that the paper makes it out to be (see comment under Originality, and the comment about Gumbel), but it still seems to represent somewhat of a departure from the conventional ways of thinking about RL algorithms. Like most new algorithmic frameworks, it seems like there may be a few kinks that have to be worked out empirically (how to choose $\eta$; is there a way of implementing the model-free version without the extra CQL term). Nonetheless, it seems like the paper could inspire more work in this direction.


**Strengths**
* The paper is generally well written, includes a thorough review of much prior work.
* The proposed framework is derived for both model-free and model-based algorithm.
* An empirical version of the proposed method is applied to to reasonable continuous control tasks. This is really nice to see in a theory paper.


**Weaknesses:**

* In a few spots, I felt like the paper claims were not entirely substantiated:
  * "One Objective to Rule Them All" in the title -- Without comparisons to "all" prior methods, I don't think this is a fair claim. Perhaps a more accurate title would be "An Objective Fusing Estimation and Planning for Exploration"
  * "algorithms predominantly undertake three tasks" -- This seems to be describing methods based on Thompson sampling and posterior sampling, but I'm not sure that R-MAX-style methods fit this mold
  * "outperforms MBPO by a stable margin ... showcases greater sample efficiency" -- I'm a bit unsure what these statements are referring to. My read of Figure 1 is that there might be statistically significant gains on 2 / 8 tasks (cheetah-vel-sparse, ant-vel-sparse). I would still vote for accepting the paper even if it only outperforms baselines on 2/8 tasks (but I think that accurately portraying the results is very important); this is especially true if we see large benefits on some tabular settings (see suggestion under Originality).
* The paper alludes to "data-dependent level-sets" in a number of places, but never defines what these are. I'd recommend either including a few sentences describing similarities/differences, or moving this discussion to the appendix.
* I have a few questions about some of the technical details (see below).
* From a practical perspective, the model-free version of the method seems to violate conventional wisdom: many practical TD methods (e.g., TD3, MBPO) choose the Q function that has lowest Q-values, whereas the proposed method chooses the one with highest Q values. Not only does this make be a bit concerned that the theoretical framework might not always work well in practice (as evinced by the need for the additional CQL regularizer), but it also makes me wonder exactly how the model-free version of the method is implemented when combined with TD3 (does it take the min or max of the Q functions)? At least for the model-free version, one thing that might help would be to sure that using the CQL regularizer allows the _same_ method to achieve good results in both online and offline settings.


---------------------

**Minor comments**:
* In the abstract, mention that this is going to be a theory paper. I expected this to be an empirical paper until partway through the intro, when I realized that all of the citations were referring to other theory papers. As such, I found the claim that prior methods "involve impractical algorithmic components" a bit strange, as prior _practical_ methods cannot require impractical components.
* L45 -- L47: I found this part a bit unclear because it was unclear what the optimization variable is (a policy, a model, data). This point is clarified later.
* L50: I found this part a bit unclear because it was unclear what the hypothesis class was over (policies, models, etc).
* Some small grammar errors throughout. An automated grammar checker should catch these (e.g., dropped articles)
* L80 -- This definition could be cut, as it's already included a few paragraphs earlier. Cutting this would make it possible to remove some of the whitespace hacks.
* L101 -- What is "BE"?
* Empirically, in many TD algorithms the TD error _increases_ as the policy improves. The standard explanation is that collecting non-zero rewards results in higher TD errors. This seems potentially related to the analysis in this paper.
* L260 -- Including a table for comparing these prior methods might help.

**Questions:**

1. How important is the value of $\eta$ for both the theoretical and empirical results?
2. L147 "Without loss of generality" -- Can the authors elaborate on this point? My guess is that we can ignore stochastic rewards because we could define a new MDP that has a copy of each state for each possible reward that that state could have (and update the transition probabilities accordingly). I'm not sure why it's OK to assume that the reward function is known.
3. There's a sense in which the proposed method resembles posterior sampling methods, by drawing a connection with Gumbel-softmax-style sampling. One way to sample a random variable is to _deterministically_ choose the value that maximizes likelihood + noise, for an appropriately chosen noise distribution (e.g., the Gumbel distribution for categorical distributions). I'm curious if we can interpret posterior sampling methods as a version of Eq. 3.1 where the first term ($V(x)$) is replaced by an appropriate noise distribution (which has nothing to do with the value).
4. L206: Does $V_f^\pi$ make sense for both the model-based and model-free versions of the method? For the model-based version, I'm interpreting this as the expected returns of policy $\pi$ under model $f$. But for the model-free version, I'm unsure if this is supposed to be the expected returns of $\pi$, or the estimate from (value function) $f$.
5. L206, "reduced to vanilla actor-critic": Is this true for the model-based version of the method?
6. Is the second term in Eq. 3.3 required?

**Limitations:**

Limitations are not discussed in the main paper. The appendix has a cursory paragraph on limitations. I would recommend including the limitations in the main text, and including a transparent discussion of benefits/limitations of the proposed method relative to prior methods. For example, the first term in Eq. 3.1 seems to sometimes cause optimization problems. Ensemble-based methods might be more stable.

---

> ### Author Rebuttal · Authors · 2023-08-08
>
> **Originality: Compare MEX with [1] on Gridworld environments.**
> Based on [3], we compare MEX on Gridworld with [1] in Figure 2 of **General Response**.
>
> **Clarity: Comparison to related works and organization of Section 5.**
>
> Please see **General Response Q3** for a brief comparison. We will add the comparison and reorganize Section 5 in the revision.
>
> **Weaknenss 1:Some of the paper claims were not entirely substantiated.**
>
> - *About the title*: By using "all", we meant to refer to the three of "Estimation", "Planning", and "Exploration", because our MEX objective can fuse all three components in a single objective.
> - *About "algorithms predominantly undertake three tasks"*: The tasks of estimation, planning, and exploration are the main components of most existing sample-efficient online RL algorithms, including but not restricted to Thompson-sampling-style methods. Essentially, all of these components are necessary for both model-free and model-based algorithms (TS-style method can also be either model-free or model-based).
> - *About "outperforms MBPO by a stable margin,..., showcases greater sample efficiency":* We clarify that the mean return of MEX-MB is higher than the MBPO baseline in the sparse-reward tasks, including cheetah-vel-sparse, ant-vel-sparse, hopper-vel-sparse, and ant-vel-sparse, especially the first two. Besides, MEX-MB outperforms MBPO with fewer samples in the standard Mujoco tasks, including walker2d, half cheetah, and ant. We will revise our statement to make it more accurate.
>
> **Weakness 2: About "data-dependent level-sets" in a number of places.**
>
> Thanks for pointing that out! We will add a concrete discussion about the "data-dependent level-sets" adopted by previous theoretical works.
>
> **Weakness 3: The model-free version seems to violate conventional wisdom: many practical methods (e.g., TD3, MBPO) choose the Q function that has lowest Q-values.**
>
> Firstly, the method to choose the lowest Q-value is **NOT** addressing the same problem that MEX wants to solve. TD3 chooses the lowest value between two separate Q-networks to handle the overestimation faced when estimating things like $E[\max_{a\in\mathcal{A}}\{Q(x,a)\}]$. In contrast, MEX chooses a high Q-value in order to incentivize exploration. These two methods do not contradict each other and can be implemented simultaneously. That is, when calculating the Bellman target, one chooses the lower value from two separate Q networks. When optimizing the Q-networks, we still use the objective (6.3) that combines TD error and Q-value.
>
> Secondly, we note that the CQL regularizer is actually a kind of entropy regularization (Appendix H.2). We use this in order to stabilize training. We highlight that the sign of the regularizer here is also the **opposite** to that adopted by the original offline CQL paper.
>
> Finally, we clarify that our method to choose a high Q-value does **NOT** violate conventional wisdom. Prevalent approaches in deep online RL that achieve good exploration commonly involve a bias to high Q-values, for instance, an ensemble of neural networks [62, 48, 13 cited in paper], intrinsic motivation-driven methods [26, 3, 10 cited in paper], etc. MEX is consistent with their convention of choosing a relatively high Q-value to help exploration.
>
> **Minor comments regarding writing and organization.**
>
> Thanks for all your detailed comments! We will try to improve the readability and organization of our paper following your suggestions point-by-point.
>
> **Question 1: How important is the value of $\eta$ for both the theoretical and empirical results?**
>
> See **General Response Q2** for an explanation.
>
> **Question 2: About the known and deterministic reward, without loss of generality.**
>
> Thanks for pointing this out! The better statement is that our result can be readily generalized to the case when the reward is stochastic and unknown. The techniques have been presented in, e.g., [2]. Thanks to these previous works, we can present our results in a simplified way.
>
> **Question 3: About using Gumbel-softmax to connect MEX to posterior sampling.**
>
> This might be correct, but it is not clear whether this could achieve the desired theoretical results if we replace the value function term with a *value-independent* noise. Also, the original Gumbel-softmax trick is operated on a finite set of choices, where for each possible choice, a Gumbel noise is added. But in our setup, we choose a function from a possibly infinite class. Thus to implement a Gumbel-softmax trick, a stochastic process indexed by $f$ is required. This is unclear and complicated in practice.
>
> **Question 4: Does $V_f^{\pi}$ make sense for both the model-based and model-free versions of the method?**
>
> We clarify this as follows. As discussed in Example 2.1, for the model-free version, we use $V_{h, f}(x)$ to refer to $\max_{a\in\mathcal{A}}f(x,a)$. However, we **did not** use the notation $V_{h,f}^{\pi}$. This notation only appears in some intermediate definitions in Example 2.2 for the model-based case and does not appear in the unified algorithm and theory. We may only use the notation $V_h^{\pi}$ to refer to the value function of $\pi$ under the true model. Please check that out in our paper.
>
> **Question 5: About the reduction to vanilla actor-critic for the model-based version of MEX.**
>
> Here we only aimed to discuss the high-level relationship between the model-free version of MEX and the vanilla actor-critic. We will make it clearer in the revision.
>
> **Question 6: Is the second term in Eqn. (3.3) required?**
>
> See to **General Response Q1** for an explanation.
>
> [1] Eysenbach, Benjamin, et al. "Mismatched no more: Joint model-policy optimization for model-based rl.".
>
> [2] Agarwal, Alekh, and Tong Zhang. "Model-based rl with optimistic posterior sampling: Structural conditions and sample complexity."
>
> [3] Wu, Chenyang, et al. "Bayesian optimistic optimization: Optimistic exploration for model-based reinforcement learning."

---

> > ### Comment · Reviewer_wpGc · 2023-08-10
> > **Reviewer response**
> >
> > Dear authors,
> >
> > Thanks for the detailed response and new experiments. This definitely helps clarify my understanding of the paper. I have one request: please revise the title to be more precise. I title that implies that the proposed method is "ruling over estimation" seems presumptuous and will likely be inaccurate in a few years.
> >
> > Also, please make sure to revise the paper to address the discussion with many reviewers about discussing the differences/similarities with prior work.
> >
> > Best,
> > Reviewer

---

> > > ### Author Response · Authors · 2023-08-10
> > > **Reply by Authors**
> > >
> > > Dear Reviewer wpGc,
> > >
> > > Thank you for your review and support. We will incorporate your valuable suggestions into our paper as we revise it based on the feedback from all reviewers. As a part of the revision, we will change the title to a more precise one and add detailed discussions with prior works. Your comments greatly assist us in strengthening the overall quality of our work.
> > >
> > > Best regards,
> > > Authors

---

### Official Review · Reviewer_vhgd · 2023-07-05

**Soundness:** 4 excellent
**Presentation:** 3 good
**Contribution:** 4 excellent
**Rating:** 8
**Confidence:** 3

**Summary:**

The paper presents a unified objective for optimizing regret in online RL algorithms with theoretical justification and experimental evaluation. The objective crucially dos not dependent on posterior estimation or bi-level optimization, making it easier to implement in practice than previous theory motivated algorithms. The authors provide a thorough theoretical analysis of their framework, covering both model-based and model-free approaches.

**Strengths:**

The paper is very well crafted, and provides an algorithm that is both theoretically sound and empirically impactful, which is always a great achievement. Even though the presentation is necessarily dense at times, given the theoretical nature of the main contribution, the paper is fairly accessible, with the core intuitions mostly presented well. The results and the practical algorithm presented in the empirical section at the end are surprisingly intuitive and seem straightforward to implement on top of several model-based RL algorithms, not just MBPO.

Caveat: I am not necessarily qualified to comment on the correctness of all mathematical details or the impact of the theoretical contributions. I am overall familiar with most concepts used (i.e. Bellman and Eluder rank) but not a core RL theory researcher, so I might be missing some details.

**Weaknesses:**

Given the theoretical nature of the paper, the presentation is necessarily dense, which makes it hard to grasp a couple of intuitive concepts. While the authors do their best to present all main components, here are some suggestions for improvement:
- The authors use generic functions several times, most prominently in Section 3, 3.2 and 3.3. They defer concrete realizations to later, but it would be helpful to provide these examples earlier so that readers unfamiliar with the exact approach have an easier time understanding the role that these function play. (Similarly in Section 5, 5.1)
- The authors use a unified notation for model-free and model-based hypotheses. While this makes generic statements easier to provide, it led to some confusion on my part because I lost track of whether a model-free or model-based hypothesis was meant at some points. Potentially a specialized notation would clarify these whenever a concrete algorithmic framework is discussed.
- While I know that this is not always the main stylistic choice in theory papers, I want to encourage the authors to explain why each assumption is used, i.e. what role they play in proofs. This is especially important for 3.1, which is very abstract and hard to relate to anything this early in the presentation.

The role of the exploration policy is not elaborated on in the concrete implementation, etc. This is promised in line 199, but I cannot find a note on the exploration policy in Section 5.

The full loss presented in 6.3 does not clarify how the additional term fulfills all the requirements introduced in Section 4. While I was able to piece together most of it, an explicit explanation here would greatly improve the ease with which the role of the assumptions can be understood. In addition, some intuitive description of what this objective optimizes would be nice to have.

The experimental section is missing a discussion on how $\eta'$ is set in practice (also, why is $\eta'$ used instead of $\eta$?).

211: for seek of -> for the sake of?

**Questions:**

Assumption 4.3 and Proposition 5: Are there requirements on the exploration policy to guarantee that sufficient coverage is obtained for the supervised learning guarantee? I might be misunderstanding the exact details here, but for the supervised guarantee to hold, it feels like some assumptions on the data is needed?

Section 5 (line 279) makes it seem like the choice of $l$ depends on characteristics of the MDP. If so, what aspects of the MDP need to be know a priori to make this choice?

Line 298: Is the choice of the Hellinger distance crucial, or would other choices work here as well?

6.2 If a differentiable model of the environment is learned, would it be possible to obtain the gradient wrt to the value function directly, without resorting to a PG style estimator? This would be directly implementable in MBPO, as the model is fully differentiable using reparametrization.

The experimental section results miss some standard information: what implementation of MBPO was used (this is also important to obtain hyperparameters), how where the environments chosen (especially why is Humanoid missing), and what is denoted by the shaded area in the plots.

Several times "specific MDP structure" is mentioned (lines201, 275, 304 [I assume here it refers to the GEC assumption?]), but the authors do not fully clarify what is meant by this. What is meant by this?

**Limitations:**

The authors do not address the limitations of their work. Potential discussion points are: the reliance on structural assumptions in the MDP are not provided in-depth, and the tuning of the parameter $\eta$.

---

> ### Author Rebuttal · Authors · 2023-08-08
>
> **Weakness 1: The presentation is necessarily dense, making it hard to grasp a couple of intuitive concepts.**
>
> Thanks for the constructive comments! We address them in the following.
>
> -  *Function approximations:*  we will present the concrete realizations of the general function approximations ealier to make the audience get familiar more easily. We will use $\mathcal{F}$ and $\mathcal{M}$ to distinguish model-free and model-based hypotheses in the revised version.
> - *Bellman completeness assumption:* Please refer to **General Response Q1** for an explanation.
>
> **Weakness 3: The role of the exploration policy is not elaborated for the concrete implementations.**
>
> Thanks for pointing it out. We actually referred to Appendix E for detailed discussions of each concrete example, including the choice of the exploration policy. We will add these to the main texts in the revision.
>
> **Weakness 4: About the loss Eqn. (6.3) used in experiments.**
>
> We will give more elaboration in the revision. Intuitively, the whole second term of Eqn. (6.3) corresponds to the exploration term $\max_{a_1\in\mathcal{A}}Q_{1,f}(x_1,a_1)$ in Sections 3 & 4, where we also used some tricks to stabilize training. In specific, motivated by CQL [1], we subtract the model value $E_{x\sim\beta}E_{a\sim\pi(\cdot|x)}[Q_{\theta} (x,a)]$ by $E_{x\sim\beta}E_{a\sim \mu}[Q_{\theta}(x,a)]$, where $\beta$ is the state distribution of the replay buffer, $Q_\theta$ is the parameterized Q-network, and $\mu$ is the uniform distribution over the actions. By introducing $E_{x\sim\beta}E_{a\sim\mu}[Q_{\theta}(x,a)]$, we can reduce the variance during optimizing $\theta$ for the model value, which stabilizes the training process.
>
> **Weakness 5: About the hyper-parameter $\eta'$ in experiment.**
>
> Please refer to **General Response Q2** for an explanation.
>
> **Question 1: About Assumption 4.3 and Proposition 5.**
>
>
> We clarify that for both model-based and model-free settings, we **do not** require any coverage assumptions on the exploration policy to obtain the supervised learning guarantee. All we considered is a purely online learning protocol. The intuition is that the expectation involved in $E_{\xi_h\sim\pi_{\exp}}$ coincides with the policy to collect the data, thus the concentration ineq. doesn't require coverage conditions.
>
> **Question 2: What aspects of the MDP need to be known a priori to make the choice of $l$?**
>
>
> We note that the technical treatment presented in line 279 is mainly due to the convention of the original GEC paper [2], where their authors mainly wanted to handle the linear mixture model in the model-free case and get a $\sqrt{T}$-regret. Indeed, for most of the time, the loss choice is natural in practice. E.g., for the model-free case, $l$ would be the TD error $f_h(x_h,a_h) - r_h - f_{h+1}(x_{h+1})$. Meanwhile, for the model-based case, the loss estimator $\ell$ would just be the log likelihood. We remark that the linear mixture model can be handled in a model-based manner and does not rely on the fact that its customized loss is known a priori.
>
> **Question 3: Is the choice of the Hellinger distance crucial, or would other choices work here as well?**
>
> We note that there exist some variants of GEC using TV norm, but we can show that GEC-Hellinger is always smaller or equal to the GEC-TV. We do not include a detailed discussion on this since the main focus of this paper is the new algorithmic framework MEX. We will add a footnote to comment on this.
>
> **Question 4: 6.2 If a differentiable model of the environment is learned, would it be possible to obtain the gradient wrt to the value function directly, without using a PG style estimator?**
>
> Yes, that would be possible. By adopting methods such that the planning process (corresponding to the optimal value function term in Mex's target) is differentiable, e.g., [3], we can obtain the gradient wrt to the value function directly in a differentiable model of the environment, e.g., MBPO.
>
> **Question 5: The experimental section results miss some standard information and add more experiments on the Humanoid environment.**
>
> We used the MBPO implementation and the hyperparameters provided in the *mbrl-lib*([4]). The experiments in the paper are conducted in both the standard and the sparse-reward Mujoco locomotion tasks. We also clarify that the shaded area in all plots in the paper corresponds to the standard deviation among five random seeds.  As for the new humanoid task, we test the performance of model-based MEX in this task and report the results in Figure 1 of **General Response**, where $\eta^\prime=1e-3$.
>
> **Question 6: The authors do not fully clarify what is meant by "specific MDP structure".**
>
> Thanks for pointing this out! We will elaborate on this in the revised version. To be specific,
>
> - Line 201: we mainly refer to the difference in Q-type and V-type problems in the literature: whether the action distribution used in the discrepancy loss $\ell$ is the same as the historical one or not. If not, we need one step of uniform exploration over the action space in the exploration policy;
> - Line 275: we refer to the transition and reward model. We mean that for different MDPs, our analysis carries on in a unified manner, insead of a case-by-case one;
> - Line 304: we mean that the MDP should have some specific structures so that the GEC condition is satisfied.
>
> **References:**
>
> [1] Kumar, Aviral, et al. "Conservative q-learning for offline reinforcement learning." Advances in Neural Information Processing Systems 33 (2020).
>
> [2] Zhong, Han, et al. "A posterior sampling framework for interactive decision making." *arXiv preprint arXiv:2211.01962* (2022).
>
> [3] Amos, Brandon, et al. "Differentiable mpc for end-to-end planning and control." Advances in neural information processing systems 31 (2018).
>
> [4] Pineda, Luis, et al. "Mbrl-lib: A modular library for model-based reinforcement learning." arXiv preprint arXiv:2104.10159 (2021).

---

> > ### Comment · Reviewer_vhgd · 2023-08-10
> > **Reply**
> >
> > Thanks for the reply. This covers all my remaining concerns. Thank you for your contribution!

---

> > > ### Author Response · Authors · 2023-08-10
> > > **Reply by Authors**
> > >
> > > Dear Reviewer vhgd,
> > >
> > > Thank you for your review and support. We will incorporate your valuable suggestions into our paper as we revise it based on the feedback from all reviewers. Your comments greatly assist us in strengthening the overall quality of our work.
> > >
> > > Best regards,
> > > Authors

---

### Official Review · Reviewer_44XZ · 2023-07-06

**Soundness:** 3 good
**Presentation:** 3 good
**Contribution:** 3 good
**Rating:** 7
**Confidence:** 4

**Summary:**

This paper proposes an RL framework, maximize to explore (MEX), that requires solving an unconstrained optimization which encompasses estimation, planning, and exploration. It proposes both model-free and model-based algorithms and further extends this framework to zero-sum Markov game setting. All these algorithms are proved to be efficient. It also designs practical algorithms based on existing deep RL baselines. Experimental results show that the proposed algorithms outperform baselines in sparse reward environments.

**Strengths:**

This paper proposes a general framework that is adapted to various settings while enjoying favorable theoretical guarantee. Its result generalizes previous ones and is important. It is technically sound, clearly written, and well organized.

**Weaknesses:**

The idea of maximizing to explore is not novel, and previous work are not adequately cited. This line of research is first developed by [Kumar & Becker (1982)](https://ieeexplore.ieee.org/document/1102878) who proposes an estimation criterion that biases maximum likelihood estimation with the cost (or value) of decision. In recent years, several work ([Liu et al., 2020](https://proceedings.mlr.press/v119/liu20g.html); [Mete et al., 2021](https://proceedings.mlr.press/v144/mete21a.html); [Hung et al., 2021](https://ojs.aaai.org/index.php/AAAI/article/view/16961); [Mete et al., 2022](https://ieeexplore.ieee.org/document/9751189/); [Mete et al., 2022](https://proceedings.neurips.cc/paper_files/paper/2022/hash/3c601cd5866099648c6dc783e7f39858-Abstract-Conference.html); [Wu et al., 2022](https://proceedings.neurips.cc/paper_files/paper/2022/hash/5bcb807ae43ad0851a6ba6162a866404-Abstract-Conference.html)) have applied this method to different settings and proved some theoretical results. In particular, [Wu et al. (2022)](https://proceedings.neurips.cc/paper_files/paper/2022/hash/5bcb807ae43ad0851a6ba6162a866404-Abstract-Conference.html) gives a theoretical result similar to the one for model-based MEX in this paper. Rather than completely ignoring previous efforts, it is more appropriate to propose MEX as a further generalization of previous research.

The proof of regret bound in this paper relies on the low generalized eluder coefficient assumption. However, it is not discussed whether this assumption hold in practice especially when the hypothesis class is parameterized by neural networks.


**Questions:**

While the theory indicates that the coefficient balancing value and evidence should change over time, it is set to be a constant in practice. Why is it the case?

**Limitations:**

This paper has not discussed its limitations, especially about optimization aspects of the learning objective and the practicality of assumptions.

---

> ### Author Rebuttal · Authors · 2023-08-08
>
>
> **Weakness 1: Lacking discussions of previous research that have similar ideas.**
>
> Thanks very much for pointing these related works out! We will add these works and discuss them in detail in the revision. In the following, we compare our work with these works briefly.
>
> [1]: While this work firstly proposed an estimation criterion that biases maximum likelihood estimation with the cost/value, their algorithm is actually different from ours, by their Eqn. (6) - (8) in Section 3, their algorithm performs the estimation of model $\alpha$ and policy optimization separately, for which they only obtained asymptotic convergence guarantees. Also, how well their decision rule explores remains unknown in theory. In contrast, MEX adopts a single optimization objective that combines estimation with policy optimization, which also ensures sample-efficient online exploration.
>
> [2,3,4,5,6]: They study Reward-Biased Maximum Likelihood Estimation (RBMLE) in Multi-arm bandit ([2]), Linear Stochastic Bandits ([3]), tabular RL ([4]), and Linear Quadratic Regulator settings (a linear parameterized model of MDP,[5,6]) and also obtain the theoretical guarantees. While these settings are special cases for our proposed algorithms and our proven theoretical guarantee can also be generalized to these concrete cases. As we claim in this paper, our main contribution is to address the exploration-exploitation trade-off issue under general function approximation, which makes our work differs from these papers.
>
>
> [7]: It's true that this work considers an algorithm similar to MEX, but our theory differs from theirs in both techniques and results. Our theory is based upon a unified framework of online RL with general function approximations, which covers their setup for model-based hypothesis with kernel function approximation (RKHS). More importantly, they derived asymptotic regret of their algorithm based upon certain uniform boundedness and asymptotic normality assumptions, which are relatively strong conditions. In contrast, we derived finite sample regret upper bound for MEX, and the only fundamental assumption needed is a lower Generalized Eluder Coefficient (GEC) MDP, which contains almost all known theoretically tractable MDP classes (therefore covers their RKHS model). Finally, our paper further extends MEX to two-player zero-sum Markov games where similar algorithms and theories are previously unknown to the best of our knowledge.
>
> Also, the works mentioned above do not design experiments in deep RL environments, while we propose deep RL implementations and demonstrate their effectiveness in MuJoco environment.
>
> **Weakness 2: It is not discussed whether the low GEC assumption hold in practice especially when the hypothesis class is parameterized by neural networks.**
>
> We note Proposition 17.20 of [8] provides a general estimation result for GEC when the function class can be embedded into an RKHS. And the original GEC paper [9] also provides many examples that are beyond linear function approximation and can capture non-linearity to some degree.
>
> **Weakness 3: It seems that the provided code for MEX-MB is wrong and is merely an implementation of MBPO.**
>
> Our code for MEX-MB is adapted from MBPO. As described in Lines 341-344, MEX-MB differs from MBPO only in the model update procedure by adding the model value gradient during model updates. The corresponding code can be found in 'MEX_MB/mbrl/models/one_dim_tr_model.py'. This also shows that our method is easy to implement with minimal computational overhead to boost performance.
>
> **Question 1: While the theory indicates that the coefficient balancing value and evidence should change over time, it is set to be a constant in practice. Why is it the case?**
>
> We note that both in the theory and experiments, we set the coefficient as a constant. In theory, we set it to be $1/\sqrt{K}$ instead of $1/\sqrt{k}$, where $K$ is the number of episodes for the entire online learning process.
>
> **References:**
>
> [1] P. Kumar and A. Becker, "A new family of optimal adaptive controllers for Markov chains," in IEEE Transactions on Automatic Control, vol. 27, no. 1, pp. 137-146, February 1982, doi: 10.1109/TAC.1982.1102878.
>
> [2] Liu, Xi, et al. "Exploration through reward biasing: Reward-biased maximum likelihood estimation for stochastic multi-armed bandits." International Conference on Machine Learning. PMLR, 2020.
>
> [3] Hung, Y.-H., Hsieh, P.-C., Liu, X., & Kumar, P. R. (2021). Reward-Biased Maximum Likelihood Estimation for Linear Stochastic Bandits. Proceedings of the AAAI Conference on Artificial Intelligence, 35(9), 7874-7882.
>
> [4] Mete, Akshay, et al. "Reward biased maximum likelihood estimation for reinforcement learning." Learning for Dynamics and Control. PMLR, 2021.
>
> [5] Mete, Akshay, Rahul Singh, and P. R. Kumar. "The RBMLE method for Reinforcement Learning." 2022 56th Annual Conference on Information Sciences and Systems (CISS). IEEE, 2022.
>
> [6] Mete, Akshay, Rahul Singh, and P. R. Kumar. "Augmented RBMLE-UCB approach for adaptive control of linear quadratic systems." Advances in Neural Information Processing Systems 35 (2022): 9302-9314.
>
> [7] Wu, Chenyang, et al. "Bayesian optimistic optimization: Optimistic exploration for model-based reinforcement learning." Advances in neural information processing systems 35 (2022): 14210-14223.
>
> [8] Zhang, Tong. Mathematical analysis of machine learning algorithms. Cambridge University Press, 2023.
>
> [9] Zhong, Han, et al. "A posterior sampling framework for interactive decision making." *arXiv preprint arXiv:2211.01962* (2022).

---

> > ### Comment · Reviewer_44XZ · 2023-08-10
> > **Re: Rebuttal by Authors**
> >
> > Thank you for your detailed and comprehensive rebuttal addressing my concerns. Your clarifications have sufficiently addressed my concerns. Taking into account your thorough rebuttal and the improvements you've outlined for your paper, I agree to raise the score from 5 to 7 and look forward to seeing the updated version of your paper.

---

> > > ### Author Response · Authors · 2023-08-10
> > > **Re:Re: Rebuttal by Authors**
> > >
> > > Dear Reviewer 44XZ,
> > >
> > > Thank you for your review and support. We will incorporate your valuable suggestions into our paper as we revise it based on the feedback from all reviewers. Your comments greatly assist us in strengthening the overall quality of our work.
> > >
> > > Best regards,
> > > Authors

---

### Author Rebuttal · Authors · 2023-08-08

**General Response:**

Thank each reviewer for the review. We provide comments on some common questions here.

**Q1: Explanations on the Bellman completeness assumption (Assumption 3.1) and the choice of loss function (3.3), especially the second term, for the model-free case.**

**A1:** **About the Bellman completeness assumption (Assumption 3.1)**: This assumption does have a very clear technical and also intuitive interpretation. It is critical for us to get a sharper supervised estimation guarantee (Assumption 4.3) for the model-free case. In short, to estimate $(\mathbb{E}[(f_{h}(x_h,a_h) - r_h - f_{h+1}(x_{h+1}))])^2$, we cannot simply use the empirical squared TD error $(f_{h}(x_h,a_h) - r_h - f_{h+1}(x_{h+1}))^2$ because $E[X^2] = (E[X])^2 + \sigma^2$, where $\sigma^2$ is the conditional variance. The main technical consideration is that with Assumption 3.1, we can use the second term in Eqn. (3.3) to control the conditional variance $\sigma^2$. The intuition why we need the function class to be complete under $\mathcal{P}_h$ is that if the $\mathcal{P}_h f$ still falls into the function class, then we can relate the conditional variance with the infimum over the function class which appears in the second term of (3.3). Otherwise, the infimum term can help nothing if $\mathcal{P}_h f$ falls outside the function class.

**About the loss function (3.3)**: At a high level, involving the second term of (3.3) is only a specific estimator construction under the Bellman completeness assumption. It is indeed used to handle the double-sampling issue via the Bellman completeness assumption to achieve a sharper supervised guarantee, as discussed above. On the other hand, in the model-free case, another straightforward choice is to use the sample mean with $m$ trajectories as the estimator to achieve a low variance, at the cost of a worse estimation guarantee (see [1] for a similar algorithmic design). From this perspective, this term is not necessary.



**Q2: About the hyper-parameter $\eta$ in theory and experiments.**

**A2:** In our theory, we select $\eta$ to be $1/\sqrt{K}$. Our theoretical result would keep the same if we set $\eta = c /\sqrt{K}$ for any constant $c>0$. Such a choice of $\eta$ is vital to balance between exploaration and exploitation so as to achieve the overall $\widetilde{\mathcal{O}}(\sqrt{K})$-regret.

In the experiments, we make some adaptations of $\eta$ to the experimental setups:

- Firstly, slightly different from the theory, in the experiments (Line 322 to 337), the empirical loss is an averaging over $(k,h)\in[K]\times[H]$ instead of a direct summation, and $\eta'$ is multiplied in front of the model value instead of the empirical loss. Thus equivalently $\eta' = 1/(\eta T)$, where $T=HK$ is the number of timesteps during training, and $\eta$ is used in the theory. This trick stabilizes the training process. Higher $\eta'$ means a higher weight on exploration, which is often used in sparse-reward settings.
- Secondly, since in the experiments the reward is not normalized to $[0,1]$ as in theory, we need to scale $\eta'$ up to a constant to match the model value and the empirical loss (we often consider squared TD error). Hence it is natural to further use $\eta'= r_{\max}/(\eta T) = r_{\max}\sqrt{K}/T$. Here $r_{\max}$ is the maximum reward in the experiment which is around $10$, $T=1e6$, and $K=1e3$. This gives $\eta'\approx 3e-3$. After some trials around $3e-3$, we choose the $\eta'$ as specified in Appendix H.3.

**Q3: Comparison with existing theoretical works.**

**A3:** We will add more explanation of the difference between MEX and previous theoretical approaches in the revision, which can better position our work. In the following, we make a brief comparison.

- Compared to the most of the version-space-based algorithms, e.g., [1, 2], our framework does not maintain a version space at each iteration and then conducts constraint optimization over the space. Therefore, our algorithm can be much more easier to be approximated in practice. We believe that such a practical computational guidance is meaningful if we want to bridge the theoretical work with the practical one;
- Compared to the posterior sampling algorithms presented in, e.g. GEC paper [3], our framework does not require the algorithms to know a good prior distribution. In practice, whether such a prior distribution required by the posterior sampling is available is not guaranteed in general.
- Compared to the E2D algorithm proposed by [4], our framework only uses an maximization oracle instead of a minimax optimization subroutine. Similarly, the E2D algorithm cannot be approximated efficiently in practice.

**References:**

[1] Du, Simon, et al. “Bilinear classes: A structural framework for provable generalization in rl.” International Conference on Machine Learning. PMLR, 2021.

[2] Jin, Chi, Qinghua Liu, and Sobhan Miryoosefi. “Bellman eluder dimension: New rich classes of rl problems, and sample-efficient algorithms.” Advances in neural information processing systems 34 (2021): 13406-13418.

[3] Zhong, Han, et al. “A posterior sampling framework for interactive decision making.” arXiv preprint arXiv:2211.01962 (2022).

[4] Foster, Dylan J., et al. "The statistical complexity of interactive decision making." arXiv preprint arXiv:2112.13487 (2021).

---

### Decision · Program_Chairs · 2023-09-21

**Decision:**

Accept (spotlight)

**Comment:**

The reviewer consensus is that this paper should be accepted; it has both seemingly novel theory and fairly interesting experimental results. There is also consensus that the title of this paper should probably be changed to make its claims more precise as this is unlikely the only objective that is relevant or needed.